# Air showers and hadronic interactions with CORSIKA 8

Maximilian Reininghaus[1] for the CORSIKA 8 Collaboration[a]

**1** Karlsruher Institut für Technologie, Institut für Astroteilchenphysik,
Postfach 3640, 76021 Karlsruhe, Germany
reininghaus@kit.edu

October 17, 2022

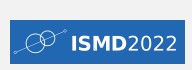

## Abstract

**The CORSIKA 8 project is a collaborative effort aiming to develop a versatile C++ framework for the simulation of extensive air showers, intended to eventually succeed the long-standing FORTRAN version. I present an overview of its current capabilities, focusing on aspects concerning the hadronic and muonic shower components. In particular, I demonstrate the "cascade lineage" feature and its application to quantify the importance of certain phase-space regions in hadronic interactions for muon production. Additionally, I show first results using Pythia 8.3, which as of late is usable as interaction model in cosmic-ray applications and is currently being integrated into CORSIKA 8.**

## 1 Introduction

A large part of the astroparticle physics community deals with the measurement of extensive air showers (EAS), particle cascades developing on macrosopically large scales of up to several 10 km. Making use of measurements of EAS observables in order to infer properties of the primary particle (high-energy cosmic rays, gamma rays, neutrinos) requires accurate predictions of these. Only Monte Caro simulations are able to provide these at the necessary level of detail, relying on numerical methods and reliable models of the physical processes involved. The phase space covered in EAS is vast: Electromagnetic (EM), weak, and strong interactions play a role; dozens of particle species are involved; energy scales range from $\sim$ keV up to ZeV. Software tools that can keep up with these requirements are a must-have and serve as a cornerstone of the field. For more than 30 years the FORTRAN code CORSIKA (Cosmic Ray Simulations for KASCADE) [1, 2] played that role and has become a de facto standard [3]. In recent years, however, it has become unfeasible to accomodate for the increasing needs of upcoming experiments by continuing to extend the existing "dinosource".

Instead, efforts have been taken to develop a new C++ code from scratch, eventually termed CORSIKA 8 [4], with the goal to provide a modern, modular and flexible framework for simulations of particle showers. Since its inception in 2018 [5], CORSIKA 8 is developed as open-source project

---

[a]full author list available at https://tinyurl.com/corsika8-202210

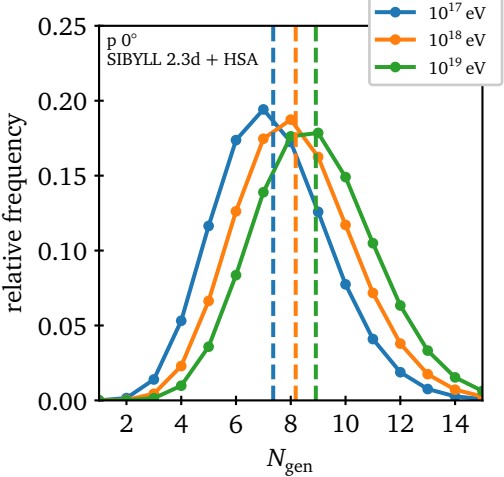 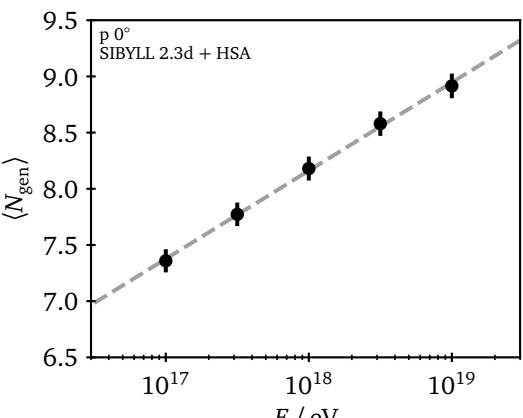

Figure 1: Left: Distributions of the number of muon ancestor generations for primary energies of $E_0 = 10^{17}, 10^{18}, 10^{19}$ eV. Vertical dashed lines indicate the mean value of the distribution of the same colour. Taken from ref. [10]. Right: Mean number of muon ancestor generations as function of primary energy. Taken from ref. [11].

by an international collaboration. The code is publicly available at the gitlab repository and usable by early adopters. At the time of writing, a large fraction of features available in the legacy version are implemented in CORSIKA 8. Moreover, CORSIKA 8 has a number of unique features that are not available in other codes.

In this article I focus on the capabilities in the hadronic and muonic sectors of EAS, which to date remain not fully understood, as a number of discrepancies between experimental data and simulations suggest [6]. More general overviews of the current state of the project can be found in refs. [7,8]. The structure of this article is as follows: In section 2 I present a study of the phase space of hadronic interactions relevant for muon production. Additionally, I show first results using Pythia 8 as hadronic interaction model in the context of air shower simulations in section 3, followed by concluding remarks.

## 2   Air shower genealogy

The muon component of EAS is a tracer of the hadronic interactions happening during the shower development. The bulk of muons observed at ground, having energies mostly in the 100 MeV to GeV range, stem from the decay of low-energy pions and kaons that form the last generation of the hadronic cascade. Measurements of the muon content of EAS induced by ultra-high energy cosmic rays (i.e., having energies $\geq 1$ EeV) performed in several experiments show that there is a significant excess of muons in data compared to simulations using up-to-date versions of hadronic interaction models [9]. It is widely believed that this discrepancy, coined *muon puzzle* [6], is the result of a lack of understanding and mismodelling of hadronic interactions.

CORSIKA 8 is particularly well suited to shed more light onto the muon puzzle: It allows keeping the complete *lineage* of particles in memory so that particles can be related to any of their predecessor generations up to the primary particle. Details on the technical implementation are

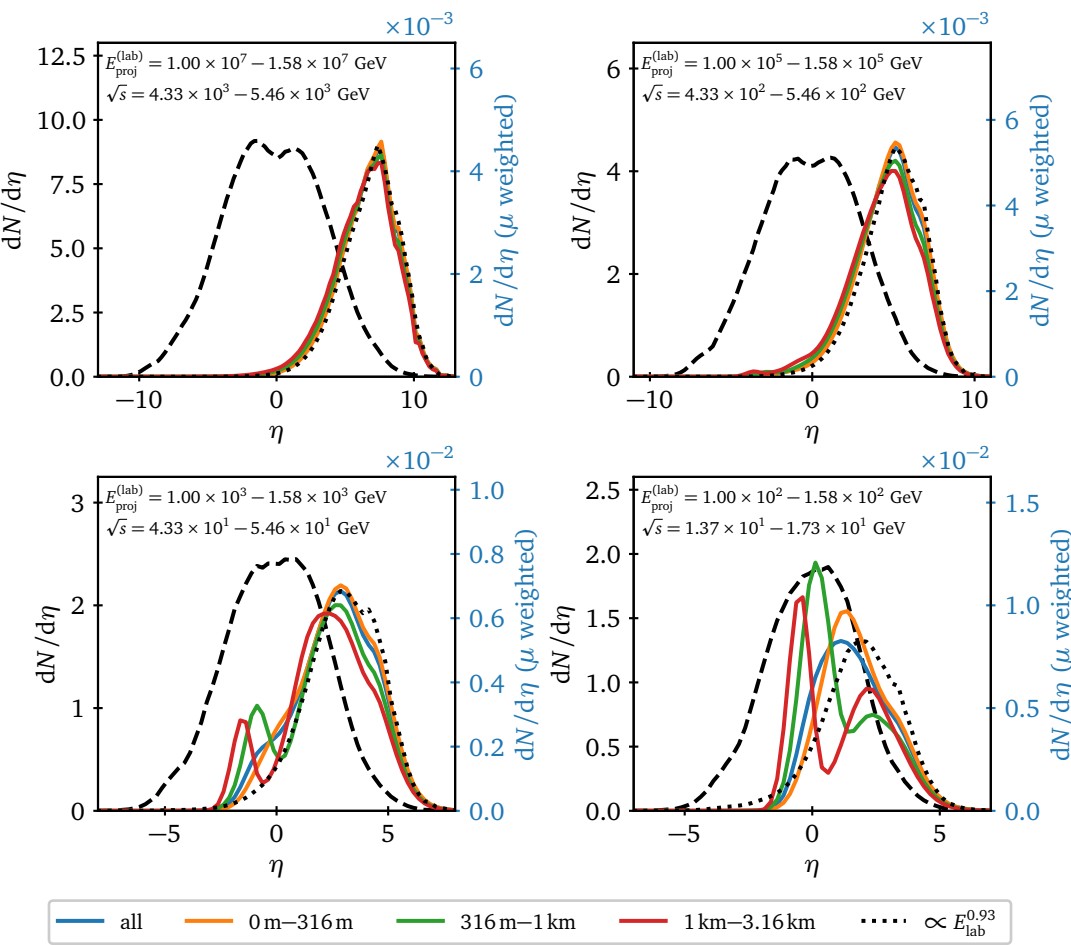

Figure 2: Pseudorapidity distributions of $\pi^\pm + \text{air} \to$ charged hadrons. The dashed line indicates generator-level distributions while the coloured and dotted lines show muon-weighted distributions of vertical, proton-induced showers of $10^{19}$ eV (in arbitrary units). Taken from ref. [10].

given in refs. [11, 12]. The first study to exploit this information is presented in ref. [10], whose results I summarize here. Figure 1 shows results regarding the number of generations $N_\text{gen}$, i.e. the number of hadronic interactions happening between the primary particle and the final muon that reaches ground. It is an important quantity because the number of muons grows exponentially with $N_\text{gen}$ and small errors in the modelling of these interactions get amplified $N_\text{gen}$ times, leading to a potentially large impact on the muon number [13]. The left plot shows the $N_\text{gen}$ distributions of proton-induced EAS with energies of $E_0 = 10^{17}, 10^{18}, 10^{19}$ eV, simulated using the interaction model SIBYLL 2.3d [14] at high energies ($> 63.1$ GeV) together with the Hillas Splitting algorithm (HSA) [15] for low energies. With increasing primary energy, the distributions shift towards higher values of $N_\text{gen}$. The right plot shows the dependence of the mean $\langle N_\text{gen} \rangle$ on the primary energy. The logarithmic behaviour follows what is expected from the Heitler–Matthews toy model [16]. A more detailed analysis is given in ref. [11].

A second study deals with the importance of different phase-space regions in hadronic interactions with respect to muon production. Figure 2 shows pseudorapidity ($\eta$) distributions of

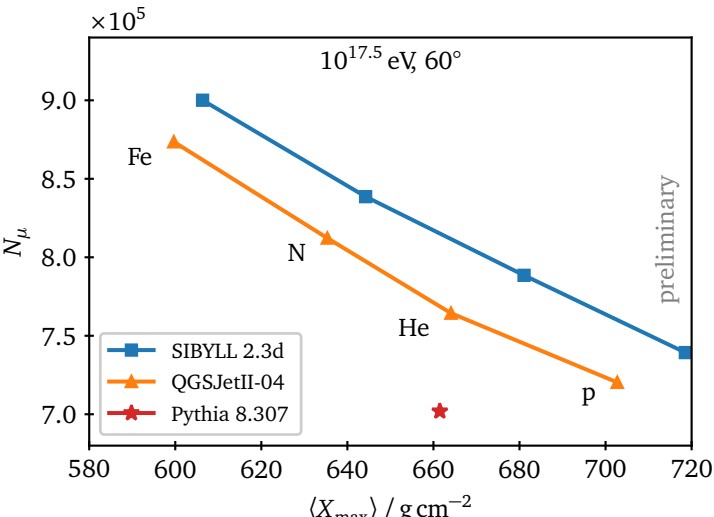

Figure 3: Mean shower maximum $\langle X_{\max} \rangle$ vs. number of muons $N_\mu$ for showers at $10^{17.5}$ eV and 60° inclination

charged hadrons in $\pi^\pm$-air collisions in four energy bins. Besides the pure generator-level spectra (dashed lines), which are in principle measurable in accelerator experiments, the *muon weighted* distributions are superimposed (coloured lines). This muon weight is given by the number of muons stemming directly or indirectly from the secondaries emitted at a given $\eta$, potentially after applying some selection criterion. The plot shows that at sufficiently high interaction energies ($\sqrt{s} \gtrsim 100$ GeV) essentially only particle production in the forward region plays a role, irrespective of the lateral distance of the muons. In this regime the muon weight can also be estimated well from the Heitler–Matthews model [6], indicated by the dotted line. At low interaction energies ($\sqrt{s} \lesssim 50$ GeV), however, the central region gains relevance especially for muons at distances larger than a few 100 m.

## 3   Towards Pythia 8 as interaction model in EAS simulations

A currently ongoing development is the inclusion of Pythia 8.3 [17] into CORSIKA 8 as hadronic interaction model. Until recently, this was technically unfeasible, but since the latest release (Pythia 8.307) a number of new features allow for an easy use in EAS simulations [18]: a) the ability to generate single events at arbitrary energies without a time-consuming re-initialization of the model, b) a much wider range of possible projectile species, c) an extended range of interaction energies down to 200 MeV (lab-frame), d) a simplified model of nuclear matter, allowing hadron-air collisions.

For a first comparison we have simulated showers with an energy of $10^{17.5}$ eV and an inclination of 60°. Hadrons and muons are fully propagated with CORSIKA 8, while EM particles are redirected into the CONEX code [19], which simulates the EM component of the shower using a numerical solution of the cascade equations describing the longitudinal development. We use QGSJet-II.04 [20], SIBYLL 2.3d and Pythia 8.307 as high-energy interaction models. In each

case Pythia is used as low-energy interaction model. Figure 3 shows the results regarding the mean shower maximum $\langle X_{\mathrm{max}} \rangle$ and $N_\mu$. While $N_\mu$ obtained with Pythia is in the same ballpark as the other models, a deviation of $\langle X_{\mathrm{max}} \rangle$ as large as the difference between proton and helium is apparent. Further studies to explain these differences, stemming from differences in hadron-air cross-sections, are ongoing and will be presented elsewhere [21]. Note that in our setup Pythia 8.307 cannot be used for nucleus-nucleus collisions at the moment so that we consider only proton-induced showers in that case.

## 4  Conclusion

CORSIKA 8 is a modern framework for the simulation of particle showers in air and other media. Over the past few years, a lot of progress has been made to make it a reliable, versatile and future-proof tool. Although not yet feature-complete, in some aspects it can be used for studies that have previously been impossible. The availability of the complete particle lineage allows detailed studies of hadronic interactions and their relevance for muon production, which helps to shed light on the muon puzzle. It emphasizes the importance of dedicated accelerator measurements at both high and low energies that cover the relevant phase-space.

The ongoing integration of Pythia 8 as new hadronic interaction model in CORSIKA 8 offers a number of new perspectives for EAS simulations. On the one hand, the reduced energy threshold of only 200 MeV renders it suitable for both low and high energies. On the other hand, Pythia 8 can be tuned by the users, allowing to study the impact of internal model parameters on EAS observables, or possibly even to conduct combined fits to accelerator measurements and EAS data at the same time.

## Acknowledgements

I thank Torbjörn Sjöstrand and Marius Utheim for providing example code that greatly simplified the integration of Pythia 8 in CORSIKA 8.

Simulations were performed on the bwForCluster BinAC of the University of Tübingen, supported by the state of Baden-Württemberg through bwHPC and the DFG through grant no. INST 37/935-1 FUGG.

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
