# Peer review of "Air showers and hadronic interactions with CORSIKA 8"

_SciPost Physics Proceedings_

## Round 1 · Referee Report · Anonymous (Referee 1) · 2022-11-25

Strengths

The report is a well-written overview of recent progress on the CORSIKA8 project and the integration of Pythia8 as a hadronic interaction model.

Weaknesses

I was missing an outlook in the final summary. The paper shows nicely what can already be done with CORSIKA 8, but it is not clear which features still need to be implemented to reach version 1.0 and feature parity with CORSIKA 7.

The tone in the introduction is a bit too informal.

Report

The paper is of high-quality and meets the acceptance criteria. I ask for a minor revision to add an outlook to the conclusions (see weaknesses).

Requested changes

p 2 “It allows keeping the” -> "It enables keeping the"

---

## Editorial Decision

resubmitted